# Biomimetic Liquid Crystal-Modified Mesoporous Silica−Based Composite Hydrogel for Soft Tissue Repair

**DOI:** 10.3390/jfb14060316

**Published:** 2023-06-08

**Authors:** Xiaoling Li, Lei Wan, Taifu Zhu, Ruiqi Li, Mu Zhang, Haibin Lu

**Affiliations:** 1Stomatological Hospital, School of Stomatology, Southern Medical University, Guangzhou 510280, China; 2The Fifth Affiliated Hospital, Southern Medical University, Guangzhou 510900, China

**Keywords:** liquid crystal, mesoporous silica nanosphere, hydrogel, angiogenesis, tissue repair

## Abstract

The reconstruction of blood vessels plays a critical role in the tissue regeneration process. However, existing wound dressings in tissue engineering face challenges due to inadequate revascularization induction and a lack of vascular structure. In this study, we report the modification of mesoporous silica nanospheres (MSNs) with liquid crystal (LC) to enhance bioactivity and biocompatibility in vitro. This LC modification facilitated crucial cellular processes such as the proliferation, migration, spreading, and expression of angiogenesis−related genes and proteins in human umbilical vein endothelial cells (HUVECs). Furthermore, we incorporated LC−modified MSN within a hydrogel matrix to create a multifunctional dressing that combines the biological benefits of LC−MSN with the mechanical advantages of a hydrogel. Upon application to full−thickness wounds, these composite hydrogels exhibited accelerated healing, evidenced by enhanced granulation tissue formation, increased collagen deposition, and improved vascular development. Our findings suggest that the LC−MSN hydrogel formulation holds significant promise for the repair and regeneration of soft tissues.

## 1. Introduction

The oral and maxillofacial region plays a pivotal role in both functional and aesthetic aspects, housing numerous vital organs and structures that serve as a “window” to society [1]. However, the maxillofacial skin, being highly exposed, is susceptible to damage from surgical procedures, accidents, and burns [2]. While natural repair processes are effective for superficial wounds, extensive, full−thickness wound healing poses significant clinical challenges, often leading to severe infections and endangering lives [3]. Therefore, there is a pressing need for a multifunctional wound dressing. Ideally, such a dressing should (1) absorb wound exudate, (2) provide protection against infection and mechanical damage, (3) exhibit excellent tissue compatibility, and, most importantly, (4) accelerate vascularization and tissue regeneration [4,5,6].

Liquid crystals (LCs) are intermediate phases of matter, also known as a mesogens, which possess microstructures between isotropic liquid and crystalline solid states [7,8]. LC phases are observed in various biological compounds, including polypeptides, lipids, nucleic acids, and polysaccharides within organisms [9]. The conformation of cell membranes closely resembles the LC state, suggesting its significance in normal physiological activities [10]. Conversely, deviations from the LC state contribute to diseases such as atherosclerosis and gallstones [11,12]. Leveraging their biomimetic properties, different LC elastomers have shown the ability to promote cell attachment, proliferation, and alignment [9,13]. In the context of wound dressing, impaired angiogenesis poses a critical limitation to soft tissue regeneration, as it hampers nutrient supply and waste removal [14,15]. While exogenous protein growth factors and therapeutic stem cells have been explored for promoting angiogenesis, challenges regarding cost and vascular endothelial growth factor (VEGF) stability in proteolytic environments persist [16,17]. Recent studies have suggested the potential of LC in promoting angiogenesis, with earlier reports demonstrating LC−induced ring formation in endothelial cells [16]. In our latest study, a multifunctional hydrogel containing LC was successfully devised which exhibited satisfactory neovascularization via providing an anisotropic viscoelastic microenvironment [18]. However, the key challenge lies in effectively delivering LC to tissues, as the direct use of polymer LCs is impractical and challenging to mold.

Mesoporous silica nanoparticles (MSNs) have garnered significant attention in the field of biomedicine due to their unique physicochemical characteristics, including a uniform pore size distribution, large specific surface area, high pore volume, tunable pore size, and favorable biocompatibility [19,20,21,22]. Over the past few decades, MSN has been extensively explored as a versatile drug delivery system for various therapeutic agents in the treatment of diseases such as cancer, diabetes, inflammation, and tissue disorders [23,24,25]. In particular, MSNs have shown promise as a carrier for osteogenesis/angiogenesis−stimulating drugs, cytokines, and diverse nanoparticles to facilitate bone tissue repair [26,27,28,29]. Thus, MSN emerges as a prospective bioactive material for tissue engineering.

Hydrogels are of great interest in the field of wound dressings due to their unique properties. With a typical three−dimensional cross−linked structure and high water content, they can create a moist environment that softens the wound surface and provides a physical barrier [30,31]. The incorporation of various functional additives, such as antibacterial agents, growth factors, nanoparticles, and magnetically responsive biomaterials, can produce binary or ternary composite multifunctional hydrogels that enhance the mechanical and biological functions of hydrogels. For instance, researchers have demonstrated that polyacrylamide−chitosan composite hydrogels can significantly enhance tensile strength and toughness and can effectively accelerate skin repair. Despite these promising results, the antibacterial performance and biocompatibility of hydrogel dressings remain suboptimal, limiting their application [32]. Further studies are needed to optimize the biocompatibility and stability of ternary composite hydrogels to ensure their safety and efficacy. Recently, some researchers have used molecular dynamics to optimize drug delivery, enabling hydrogels to achieve better physical properties and more precise drug release [33,34]. Among the naturally derived hydrogels, polysaccharides such as hyaluronic acid and chitosan (CS) are commonly employed. CS in particular is highly favored due to its good biocompatibility and degradability as well as its antibacterial and anti−inflammatory properties, which can accelerate the wound healing process. The combination of chitosan hydrogels with excellent physical properties and biomaterials with angiogenic properties holds significant promise for promoting tissue regeneration.

In this study, our primary objective was to propose a convenient and effective method for enhancing the angiogenic activity of MSN through the bioactive modification of LC. To achieve this, MSN and cholesteric LC were separately prepared and subsequently combined through ontology polymerization, resulting in a novel bionic composite material (Figure 1). Systematic characterization was conducted to optimize the morphology and relevant physical properties of both the LC and the LC−MSN composite system. In addition, the cell affinity and angiogenic capacity of the LC−MSN material were investigated for human umbilical vein endothelial cells (HUVECs). Based on the great cytocompatibility and exciting angiogenesis results in vitro, LC−MSN was processed with CS into hydrogel form for wound healing. The present study aims to develop a novel dressing that induces angiogenesis and promotes tissue regeneration in full−thickness skin wound models.

## 2. Materials and Methods

### 2.1. Chemicals and Materials

Cholesterol (degree of deacetylation ≈ 95%) was sourced from Aladdin Biochemical Technology Co., Ltd. (Shanghai, China). Undecylenic acid, chloroplatinic acid (H_2_PtCl_6_·6H_2_O), and polymethylhydrogensiloxane (PMHS) were obtained from Shanghai Macklin Biochemical Co. (Shanghai, China). Cetyltrimethylammonium bromide (CTAB), tetraethoxysilane (TEOS), azobisisobutyronitrile (AIBN), methyl acetoacetate (MAA), ethylene dimethacrylate (EDMA), and all other analytical reagents used in the experiments were procured from Tianjin Damao Chemical Reagent Factory.

### 2.2. Synthesis of LC

The detailed experimental method for LC synthesis has been previously described in the literature [8]. In brief, cholesterol and undecylenic acid underwent a simple esterification reaction to produce the undecylcholesteryl ester monomer LC. Subsequently, the side−chain polymer LC (P−UChol) was obtained through hydrosilylation of the aforementioned monomer with PMHS in toluene.

### 2.3. Synthesis of MSN and LC−Modified MSN (LC−MSN)

The classical Stoeber strategy [35] was employed for the synthesis of MSN. Ammonium hydroxide and CTAB were added to distilled water and stirred at 40 °C for 1 h. Next, TEOS dissolved in ethanol was added to the above aqueous solution and stirred at 40 °C for 5 h. In this study, the complexes based on liquid crystals and MSN were prepared using the ontology polymerization method. Initially, MSN, LC, MPDE, and AIBN were weighed in a synthesis vial, followed by the addition of acetonitrile and toluene, which were dissolved through ultrasound. Subsequently, MAA and EDMA were added and further sonicated for mixing. The mixture was then subjected to nitrogen gas to remove dissolved oxygen and reacted in a water bath at 53 °C for 4 h. After completion of the reaction, the synthesized materials were eluted with methanol/acetic acid (*v*/*v*, 9/1) using a Soxhlet extractor for 72 h until no template molecules or unreacted monomers were detected by UV spectrophotometry. The remaining unreacted substances were washed off with methanol, and the compound was dried under vacuum condition. Subsequently, a systematic examination of the characterization and physicochemical properties of LC/LC−MSN was conducted as follows.

The chemical structure of composites was analyzed by attenuated total reflectance−Fourier transformation infrared (ATR−FTIR, Bruker Tensor27, Karlsruhe, Germany). The samples were mixed with potassium bromide then pressed into discs and measured in the wavelength range 4000–400 cm^−1^.

The intermolecular and intramolecular interactions were recorded with an NMR spectrometer (Bruker, AVANCEIII400MHz, Karlsruhe, Germany). The samples were fully dissolved in deuterated chloroform and transferred to the NMR tube for scanning.

The morphology and structure of MSN and LC−MSN were characterized by transmission electron microscopy (TEM, Philips CM−120, Amsterdam, The Netherlands) at an accelerating voltage of 3 kV on a copper platform under vacuum condition.

The surface area and pore size were evaluated by the automated surface area and pore size analyzer (Quantachrome Autosorb−iQ−2, Boynton Beach, FL, USA). The surface areas were obtained by the Brunauer–Emmet–Teller (BET) method, and the pore size distributions were estimated by the Barrett–Joyner–Halenda (BJH) method. The hydrodynamic size of the samples was determined using a dynamic light scattering analyzer (DLS, SZ−100−Z, Horiba Ltd., Kyoto, Japan). The surface charges of MSN and LC−MSN in distilled water were detected by Zeta−potential (Malvern Panalytical Zetasizer Nano ZS90, Worcester, UK). The sample suspension was ultrasonically dispersed well prior to analysis. The clean point and the liquid crystal phase transition temperature of LC−MSN were measured by differential scanning calorimetry (DSC, Netzsch STA449F5, Selb, Germany) at a temperature range of 0–200 °C with a heating rate of 10 °C min^−1^ under nitrogen protection.

The texture of optical anisotropic LC−MSN composites was viewed and recorded with polarizing optical microscopy (POM, Wetzlar Leica DMRX, Wetzlar, Germany) with a heating stage. A small amount of the sample was placed on a slide, pressed firmly with a cover glass, and observed on a hot platform at room temperature.

### 2.4. Fabrication and Characterization of CS/LC−MSN Composite Hydrogel

CS was dispersed in HCl for amino protonation, after which 1, 2−propylene glycol in an amount equal to HCl was added and stirred for 1 day to obtain a homogeneous 2 wt% CS solution. The resultant LC−MSN (0.25, 0.50, 0.75, or 1.0%) was dissolved with alcohol and then dispersed into the aforementioned CS solution by equal volume under rapid stirring. The solution was poured into a mold followed by dehydration at 50 °C to obtain CS/LC−MSN alcogels. These alcogels were then immersed in NaOH for 1 day at a concentration of 1 mol/L and then washed thoroughly with phosphate buffer saline (PBS) for purification. The XRD patterns of the lyophilized CS and CS/LC−MSN specimens were acquired by using a Bruker D8−Advance powder diffract meter (Karlsruhe, Germany) equipped with Cu radiation. X−ray photoelectron spectroscopy (XPS) spectra were obtained using a Thermo Scientific ESCALAB 250 XI. The microstructures of the composite hydrogel samples were observed via scanning electron microscopy (SEM, LEO1530 VP, Philips, Amsterdam, The Netherlands). The compressive test of hydrogel samples was conducted with a universal mechanical test machine (AGI−1, Shimadzu, Kyoto, Japan).

To investigate biodegradation, the CS/LC−MSN hydrogel samples were cut into discs of approximately equal mass (500 ± 10 mg) and incubated in simulated body fluid (SBF) at 37 °C. At 1, 3, 7, 10, and 14 days, the accumulated silica ion concentrations released from composites wre detected using the inductively coupled plasma optical emission spectrometer (ICP−AES, iCAP 6300Duo, Thermo Fisher Scientific, Waltham, MA, USA). Subsequently, the discs were removed, gently washed with deionized water, dried, and observed for morphological changes using TEM during the degradation process.

### 2.5. Cellular Behavior on LC−MSN In Vitro

#### 2.5.1. Culture of the Human Umbilical Vein Endothelial Cells

Human umbilical vein endothelial cells (HUVECs; Cat. No. iCell−h110, iCell, Shanghai, China) were utilized to assess biocompatibility, cell migration, and angiogenesis capacity. The HUVECs were cultured in endothelial cell medium (ECM; Cat. No. 1001, Sciencell, Carlsbad, CA, USA) without glutamine (ECM−NG), supplemented with 5 vol.% FBS and 1 vol.% endothelial cell growth supplements (ECGs). Cells at passages 3–5 were employed for the cell experiments, and they were maintained in a humidified incubator at 37 °C with 5% CO_2_.

#### 2.5.2. Cell Proliferation Assay

The biocompatibility of LC−MSN was evaluated using a Cell Counting Kit−8 (CCK−8, Dojindo, Japan) assay. HUVECs were seeded in 96−well plates (Jet Biofil, China) at a density of 5000 cells per well. After overnight incubation, the ECM culture medium was replaced with various dilutions of CS/LC−MSN (LC−MSN concentrations: 0, 4.7, 9.4, 18.8, 37.5, 75, 150 μg/mL). The cells were further cultured for 12, 24, and 48 h, respectively. At each time point, the residual medium was removed, and the cells were washed twice with PBS. A CCK−8 working solution, prepared by mixing the CCK−8 solution and fresh medium at a volume ratio of 1:10, was added to each well. The cells were then incubated for 2 h at 37 °C. Finally, the absorbance at 450 nm for each well was measured using an enzyme−linked immunoassay (Bio−rad iMark, Hercules, CA, USA). Each group consisted of five parallel samples.

#### 2.5.3. Live/Dead Assay

To assess the cytotoxicity of LC−MSN on HUVECs, a live/dead assay was conducted using calcein AM/PI staining. HUVECs were seeded on 12−well plates at a density of 5 × 10^4^ cells/well. After incubation for 24 and 48 h, the medium was removed, and the cells were washed twice with PBS. Subsequently, calcein AM/PI reagent (Meilunbio, Dalian, China) was added to the wells following the protocol and incubated at 37 °C for 30 min in the dark. The images were then observed using an inverted fluorescence microscope (Leica DMi8, Wetzlar, Germany). Finally, the living and dead cells were counted using ImageJ software (V1.8.0.112, Media Cybernetics, Rockville, MD, USA).

#### 2.5.4. Scratch Migration Assay

The scratch assay was performed to evaluate the effect of LC−MSN on endothelial cell migration. HUVECs were seeded in 6−well plates at a density of 6 × 10^5^ cells/well and cultured in an incubator. After 24 h of incubation, a confluent monolayer of cells was formed. Two parallel scratches were created vertically across the center of the well using a 1000 μL pipette tip. The wells were then rinsed gently three times with PBS to remove free cells. Next, 2000 μL of culture medium containing LC−MSN at concentrations of 0, 9.4, 18.8, and 37.5 μg/mL was added. After incubation for 0, 24, and 48 h, images were captured using a bright−field microscope at 10× magnification (Leica DMIL−LED, Wetzlar, Germany), and the unrecovered area was calculated using ImageJ software. The experiments were performed in triplicate. The simulated wound closure was quantified using the following equation:Wound area closure (%) = (S_0_ − S_t_)/S_0_ × 100% (1)
where S_0_ represents the original area of the scratch, and S_t_ represents the area at the observation times.

#### 2.5.5. Cell Morphology

The cellular morphology of HUVECs was observed using F−actin cytoskeleton stains. After 24 and 48 h of incubation, cells were rinsed with PBS and fixed in 4% paraformaldehyde (Meilunbio, Dalian, China) for 20 min at room temperature. Subsequently, the cells were permeabilized with PBS containing 0.1% Triton X−100 (Beyotime, Shanghai, China). Intracellular F−actin filaments were then stained for 1 h using phalloidin conjugated to TRITC (1:200 dilution in PBS, Solarbio, Beijing, China). Additionally, the nuclei were stained with 300 nM DAPI (Solarbio, Beijing, China) for 30 s. After rinsing, the cell morphology was visualized using a confocal laser scanning microscope (CLSM, Leica STELLARIS5, Wetzlar, Germany).

### 2.6. Angiogenesis Assay on LC−MSN In Vitro

#### 2.6.1. Tube Formation Assay

To evaluate the impact of LC−MSN on the angiogenesis function of HUVECs, a tube formation assay was conducted. Firstly, 10 μL of BD Matrigel TM (Corning, NY, USA) was transferred to wells of the ibidi μ−plate and incubated at 37 °C for 1 h to form a gel substrate. Subsequently, HUVECs cultured in ECM or treated with LC−MSN for 24 or 48 h were digested and resuspended in ECM at a density of 1 × 10^5^ cells/mL, respectively. Then, 50 μL of the cell suspension was added onto the Matrigel. After 6 h of culture, the cells on the Matrigel were observed using an optical microscope (Leica DMIL−LED, Wetzlar, Germany) under bright field illumination, and five random microscopic areas were selected for photography. The obtained images were processed using ImageJ software, and the meshes, nodes, and total length formed by the cells were counted.

#### 2.6.2. qPCR

The expression of angiogenesis−related genes in HUVECs was assessed using qPCR. Cells were seeded in a 6−well plate at a density of 5 × 10^5^ cells per well and cultured in ECM or the medium containing LC−MSN for 48 h. Total RNAs of the cells in each group were extracted using the TRIzol reagent (Invitrogen, Carlsbad, CA, USA) and reverse−transcribed into cDNA using an Evo M−MLV RT Premix kit (Accurate Biology, Changsha, China). RT−qPCR was performed using the SYBR Green QPCR Master Mix (APE × BIO, Houston, TX, USA) with a real−time qPCR system (Roche LightCycler−96, Switzerland) for the following genes: CD31, VEGF, ANG, and GAPDH. GAPDH was used as the endogenous control. The expression levels of the target genes were calculated and normalized using the 2^−ΔΔCt^ method. The primer sequences are listed in Table 1, and there were three parallel samples in each group.

#### 2.6.3. Western Blot

The expression of angiogenesis−related proteins in HUVECs was analyzed using western blot. The cell seeding and treatment procedures were the same as those for qPCR. After 48 h, the cells were lysed on ice for 30 min with RIPA lysis buffer containing a protease inhibitor. The lysate was then centrifuged at 12,000× *g* for 15 min at 4 °C, and the supernatant was collected. The protein concentration was determined using a BCA protein assay kit (GLPBIO, Montclair, CA, USA). Loading buffer (NCM, Suzhou, China) was added to the samples at a ratio of 5:1, followed by heating at 100 °C for 10 min. Subsequently, 20 μg of the samples was separated by polyacrylamide gel electrophoresis and transferred to a PVDF membrane (Merck Millipore, Darmstadt, Germany). The membrane was blocked with 5% BSA for 30 min and then incubated with primary antibodies against CD31 (Abcam, ab76533), VEGF (Abcam, ab46154), ANG (Abcam, ab276132), or GAPDH (MBL, M171−7) at 4 °C overnight. After washing the membrane with TBST three times, it was incubated with a horseradish peroxidase (HRP)−conjugated secondary antibody at room temperature for 2 h. The protein bands were visualized using chemiluminescence with the ECL detection kit (NCM, Suzhou, China), and the intensity of the bands was analyzed using ImageJ software. All experiments were performed with three replicates.

### 2.7. Animal Study

#### 2.7.1. In Vivo Wound Healing

A total of fifteen healthy Sprague–Dawley rats (7–8 weeks old) were included in the approved in vivo study, authorized by the Animal Ethics Committee of Southern Medical University under approval/accreditation number SCXK(YUE)2021−0041. To assess the effect of the LC−MSN−incorporated hydrogel on wound healing, three subgroups were established: the control, CS hydrogel, and CS/LC−MSN hydrogel groups. The rats were anesthetized by intraperitoneal injection of pentobarbital sodium (3% *w*/*v*, 40 mg/mL). Following shaving, four full−thickness skin wounds (round, diameter = 6 mm) were created symmetrically on the dorsal skin using a biopsy punch. The hydrogels were then applied to the wounds, while the control wounds remained untreated but were covered with Tegaderm transparent dressing (3M Health Care, Germany) to prevent infection. The healing progress of each wound was observed and photographed at 0, 3, 7, and 14 days. The degree of wound closure was quantitatively determined by measuring the wound area using ImageJ software and calculated using the following equation:Wound healing rate (%) = (A_0_ − A_t_)/A_0_ × 100%(2)
where A_0_ represents the initial wound area and A_t_ is the wound area at the specific time points (0, 3, 7, and 14 days). At the end of the experiment, the rats were euthanized using an isoflurane overdose. The wound sites were excised for subsequent histological and immunohistochemistry analyses.

#### 2.7.2. Histological Analysis

Wound tissue samples were collected for histological evaluation. The samples were fixed in 4% paraformaldehyde, dehydrated through an ethanol series, embedded in paraffin, and then sectioned into 5 µm thick slices using a microtome. The paraffin sections were mounted on slides and subjected to staining with hematoxylin−eosin (H&E) and Masson’s trichrome stains according to the manufacturer’s instructions. These stains allowed for the examination of regenerated tissue and collagen deposition, respectively. Digital images of all images of all sections were captured using a digital pathology slide scanner (Leica, Aperio VERSA, Wetzlar, Germany).

#### 2.7.3. Immunohistochemistry Staining

Immunohistochemistry staining was performed to evaluate neovascularization and inflammation levels. Paraffin−embedded tissue sections were dewaxed, rehydrated, and then blocked with 5% bovine serum albumin for 1 h. Next, the sections were incubated overnight at 4 °C with primary antibodies: CD31 (ab76533, Abcam), interleukin−1beta (IL−1β) (ab5694, Abcam), and interleukin−6 (IL−6) (ab2105, Abcam). After three washes with PBS, the sections were incubated with a goat−anti−rabbit secondary antibody solution for 30 min. All slides were examined using a light microscope (Leica DMi8, Wetzlar, Germany).

### 2.8. Statistical Analysis

Each experiment was conducted with at least three parallel samples. Data analysis was performed using GraphPad Prism 8 (GraphPad Software, San Diego, CA, USA). The results are presented as the mean ± standard deviation (SD). Statistical differences between groups were evaluated using one−way analysis of variance and Student’s *t*−test. A *p*−value less than 0.05 was considered statistically significant.

## 3. Results

### 3.1. Synthesis and Characterization of LC−MSN

The successful modification of LC on MSN was confirmed through ATR−FTIR analysis (Figure 1A). The spectrum of LC−MSN exhibited two new peaks at 1724 cm^−1^ and 1640 cm^−1^, indicating the completion of the esterification reaction, a crucial step in LC preparation. Furthermore, the ^1^H−NMR spectrum showed new resonances at approximately 6.4 and 6.1 ppm (a, b), attributed to the C=C bonds (Figure 1B). Additionally, a signal at around 4.60 ppm (c) indicated the characteristic hydrogen on the ring carbon bound to the newly formed C=O. Therefore, the consistent results from ATR−FTIR and ^1^H−NMR confirmed that the synthesized LC−MSN conformed to the expected molecular design [8,36,37].

The morphology of MSN and LC−MSN was observed using TEM, revealing that LC−MSN maintained a spherical shape with uniform particle size and good mono−dispersibility, similar to MSN (Figure 1C). These findings suggested that LC modification did not significantly affect the average particle size and dispersion of MSN.

The N_2_ adsorption/desorption isotherms of MSN exhibited a typical type IV isotherm, indicating its mesoporous nature (Figure 1D). The average BET surface area and pore diameter of MSN were measured as 609.6 m^2^/g and 5.29 nm, respectively. After loading LC, these values decreased to 349.4 m^2^/g and 4.21 nm, respectively (Figure 1E).

To further characterize the nanoscale structure changes of MSN, DLS and Zeta potential measurements were conducted. DLS analysis revealed that the average size of MSN was 89.6 ± 5.7 nm, with a slight increase of 6.3 nm upon the addition of LC (Figure 1F). Moreover, the polymer dispersity index (PDI) in the aqueous phase was less than 0.3, indicating good dispersion stability of the nanocomposite. The Zeta potential of MSN was −38.9 ± −3.1 mV, which changed to −20.1 ± 2.1 mV upon LC incorporation (Figure 1G), suggesting the presence of a positively charged carbonyl group.

DSC thermal analysis revealed an additional endothermic peak at approximately 120 °C, corresponding to the glass transition temperature (Tg) of LC−MSN, indicating the crystalline properties of LC (Figure 1H). According to the POM images, the LC−MSN showed the typical cholesteric texture and color of anisotropic materials under a concentration of 2.5% in contrast to the MSN, and the optical field of LC−MSN under POM gradually became brighter with more distinct optical textures (Figure 1I). These results confirmed that the modification of LC into LC−MSN did not affect the properties of the LC phase.

### 3.2. Morphology and Biodegradation of CS/LC−MSN Composite Hydrogel

Considering the excellent biodegradability, biocompatibility, antibacterial, and anti−inflammatory properties of natural CS [5], the CS was used as a matrix polymer to prepare composite hydrogel scaffolds. In this work, the LC−MSN was incorporated into the CS polymer (the final concentration was 1.0%), and the CS/LC−MSN composite hydrogel system containing 0.25, 0.50, 0.75, or 1.0% of LC−MSN was fabricated by the electrostatic and intermolecular hydrogen bonding. We evaluated the mechanical properties of the different hydrogel samples through a compression test. Appendix A displays the compressive strength of the composite hydrogel enhancement with the increased concentration of LC−MSN added. Moreover, the composite scaffolds showed a decline in ductility with the introduction of 1.0% LC−MSN, despite high compressive stress. As a result, CS/LC−MSN containing 0.75% of LC−MSN was deemed to be the ideal component for the studies that followed.

The microstructure of the cross−section of CS or CS/LC−MSN hydrogels was observed by SEM. As revealed in Figure 2A,B, both the CS and CS/LC−MSN scaffolds showed a typical 3D porous morphology, while the CS/LC−MSN sample displayed interconnected porous structures with a denser, cross−linked network compared to that of the neat CS hydrogel due to the incorporation of LC−MSN, leading to the enhancement of the CS polymer network. The FTIR, XRD, and XPS analyses were further exploited to determine the chemical structure and interactions of CS/LC−MSN. Figure 2C shows the FTIR spectra of the CS and CS/LC−MSN hydrogels. The characteristic absorption peaks of −OH and N−H for LC−MSN were observed at 3380 cm^−1^, while the two absorption peaks appearing at positions 1627 and 1525 cm^−1^ corresponded to the stretching vibrations of the amide I and amide II bands in the CS chain. As for the inconspicuous characteristic peak at 2610 cm^−1^ and the broad peak at 1000–1100 cm^−1^, they belong to the Si−O−Si multiple absorption peaks of LC−MSN [38]. Further XPS survey spectra of the CS/LC−MSN composite hydrogels also confirmed the above findings (Appendix A). These outcomes demonstrated the successful formation of the CS/LC−MSN composite hydrogel.

From the XRD patterns of the CS and CS/LC−MSN specimens (Appendix A), the diffraction peaks of LC−MSN were observed around 16.7 and 19.3, which corresponded to the (200)/(110) and (203) planes of the liquid crystal [39], respectively. The biomineralization ability of the CS/LC−MSN hydrogels was evaluated in a 1.5 SBF solution. Appendix A showed that the number and size of minerals on the CS/LC−MSN scaffolds increasingly grew as the mineralization time prolonged, indicating the favorable mineralization induction performance in 1.5 SBF.

In addition, to determine the biodegradation of hydrogels, hydrogels with LC−MSN were incubated in SBF. The degradation analysis (i.e., accumulated degraded Si) demonstrated that the degradation profile of the CS/LC−MSN hydrogel was in an acceptable range of 45 to 89% during the 14 days (Figure 2E), revealing the favorable degradation ability of the composite scaffolds in vitro. It is noted that CS/LC−MSN degrades more slowly than pristine CS due to its higher cross−linking density than the CS hydrogel. The degradation rate of hydrogels depends on the density, mass, and hydrophilicity behavior of the polymers, as well as the number of linkage bonds.

To get more insights into the release of LC−MSN from composite hydrogels, the morphological changes were observed by TEM during the degradation process. The morphology of LC−MSN was observed at higher magnification, confirming the retention of its spherical shape and uniform morphology. Upon incubation, the assemblies underwent significant swelling after 1 day, leading to a loosening of the compact network structure and increased swelling over the next 14 days. Throughout this process, the amount of unreleased LC−MSN within the hydrogel gradually decreased, while maintaining good morphology and dispersion.

### 3.3. Cytocompatibility of CS/LC−MSN In Vitro

To investigate the hypothesis that the LC−MSN composite creates a favorable microenvironment for skin regeneration and cell proliferation, various assays were performed, including live/dead staining, wound scratch testing, and a cell morphology assay.

To evaluate cell proliferation, HUVECs treated with different groups were examined for up to 48 h using a CCK−8 assay. The results, shown in Figure 3C, demonstrate that LC−MSN exhibited similar cytocompatibility to the normal complete culture medium (control) in terms of HUVEC proliferation, except at concentrations higher than 75 μg/mL, which inhibited HUVEC growth on day 2. Considering that biocompatibility is essential for biomaterials intended for tissue regeneration, concentrations higher than 75 μg/mL were not used in subsequent experiments. The cytotoxicity of HUVECs was further assessed using a calcein−AM/PI kit, which stained live cells green and dead cells red. The results of live/dead staining (Figure 3A,B) aligned with the cell proliferation assay, indicating that low concentrations of LC−MSN did not affect HUVEC viability. The fluorescent intensity of the LC−MSN groups remained relatively unchanged after 24 and 48 h of incubation compared to the control group.

Next, we performed a scratch assay to examine the migration of endothelial cells from the edge to the center area, which is a critical step in the angiogenesis process. The migration behavior of each group was captured in images (Figure 3E,F). After 48 h of culturing, distinct cell migration patterns were observed in the experimental groups. The wound closure in the normal culture medium and LC−MSN at 18.8 μg/mL reached 75% and 80%, respectively. Furthermore, LC−MSN at 9.4 μg/mL and 37.5 μg/mL did not exhibit any migration inhibition.

Finally, since cell migration ability is integrally linked to the formed stress fiber and microfilament containing actin, the cellular morphology of HUVECs was observed. In comparison to the control group, the LC−MSN group at 18.8 μg/mL displayed more prominent stress fibers, actin−containing microfilaments, and fully expanded cytoskeletons (Figure 3D). Importantly, the cells exhibited well−elongated and distributed morphology, with filopodia extension and cellular spreading fronts. These observations provide a clear explanation for the beneficial effects of LC−MSN on the interconnection and migration of HUVECs. In this regard, it is speculated that the flow and order of LC helped cells to guide the extension of the cellular pseudopods. Based on the obtained results, LC−MSN at low concentrations showed no inhibition of cytocompatibility, while 18.8 μg/mL exhibited the most effective promotion of HUVEC migration.

### 3.4. Angiogenesis Assay of HUVECs In Vitro

In order to assess the angiogenic potential of LC−MSN, a tube formation assay was conducted as it is a crucial step in angiogenesis [40]. Representative images revealed that HUVECs cultured with LC−MSN at concentrations of 9.4 μg/mL and 18.8 μg/mL exhibited a higher tendency to self−assemble into capillary−like structures on Matrigel compared to those cultured with ECM alone (Figure 4A). Quantitative analysis was performed to evaluate the node number, mesh number, and total length (Figure 4B). The average numbers of nodes and meshes were significantly higher in the LC−MSN groups compared to the control group (*p* < 0.001), particularly in the 18.8 μg/mL treatment group, which exhibited nearly twice the number of nodes as the control group (*p* < 0.0001). Additionally, HUVECs stimulated with LC−MSN at 18.8 μg/mL showed a slight increase in total length (*p* < 0.001). Given that angiogenesis plays a crucial role in tissue regeneration, LC−MSN at 18.8 μg/mL demonstrated a potential advantage in wound healing.

To gain further insight into the angiogenic properties of LC−MSN, qRT−PCR and Western blot analysis were conducted. Following coculturing with LC−MSN for 48 h, the expression of several angiogenesis−related genes, including CD31, VEGF, and ANG, in HUVECs (Figure 4C) showed a significant upregulation compared to that of the control groups (*p* < 0.01). The Western blot analysis results (Figure 4D) were consistent with the qRT−PCR findings (*p* < 0.01). Collectively, these results indicate that LC−MSN at an appropriate concentration can enhance the proliferation, migration, and angiogenic ability of HUVECs, thus contributing to skin regeneration.

### 3.5. In Vivo Skin Regeneration of Full−Thickness Wounds

To evaluate the therapeutic efficacy of the LC−MSN hydrogels on soft tissue repair, full−thickness skin defects were developed on rats. The LC−MSN hydrogel−treated group demonstrated a significantly higher percentage of wound closure compared to both the blank hydrogel−treated and non−treated groups in the in vivo study of full−thickness wounds. By day 14, the LC−MSN hydrogel−treated wounds exhibited superior healing compared to the non−treated group, with almost complete wound repair in the composite gel−treated group, while approximately 3% wound closure remained in the control groups (Figure 5A,B). The histological analyses by H&E staining and Masson’s trichrome staining were further conducted after 7 or 14 days of treatment. The histological analysis using H&E staining revealed that wounds treated with the LC−MSN hydrogel displayed enhanced granulation tissue formation compared to the control groups (Figure 5C). Specifically, at 14 days post−operation, the composite gel−treated group exhibited a well−healed epithelial layer with a tightly regenerated dermis underneath, containing ample appendages. Masson’s trichrome staining further demonstrated increased collagen deposition and more organized fiber alignment, resembling the density of normal skin in the LC−MSN hydrogel−treated group (Figure 5D).

Notably, the control group exhibited a presence of injured epithelialized tissue after 14 days of treatment, along with evident dermal congestion and significant inflammation. In contrast, the CS/LC−MSN hydrogel−treated group displayed the formation of complete epithelial and dermal structures, abundant skin appendages in the healed tissue, and noticeably thicker granulation compared to the CS and control groups (Figure 5E). These findings indicate that the CS/LC−MSN composites, when used as a novel wound dressing, can accelerate epidermal remodeling and promote tissue repair.

Immunohistochemical staining of IL−1β (early inflammatory cytokine markers) and IL−6 (intermediate inflammatory cytokine markers) was performed to evaluate the inflammatory response during the cutaneous wound healing process. The goal was to further investigate the mechanisms by which CS/LC−MSN promotes wound healing. As anticipated, the CS− and CS/LC−MSN−treated groups exhibited a lower expression of IL−1β or IL−6 (brown staining) in the wound tissue compared to the control groups (Figure 6). This This demonstrates that CS significantly inhibits the inflammatory reaction at the wound site.

Furthermore, at day 7 post−treatment, neovascularization around the wound was assessed through the immunohistochemical staining of CD31. The LC−MSN hydrogel−treated group displayed a significantly higher density of blood vessels with larger diameters and thicker walls compared to the other groups (Figure 6), indicating the formation of new blood vessels. Although the blank hydrogel group showed a slight increase in the number of blood vessels compared to the control group, the difference was not significant. Hydrogels loaded with LC−MSN demonstrated favorable properties in promoting vascularization during full−thickness skin repair. Consequently, the resulting CS/LC−MSN composite scaffolds exert a dual function of inhibiting the inflammatory microenvironment and promoting angiogenesis, thereby exhibiting synergistic effects in triggering skin regeneration.

## 4. Discussion

Injuries to the oral and maxillofacial tissues resulting from burns, surgery, and trauma can compromise their protective barrier function and impede proper healing. The promotion of neovascularization by endothelial cells at the wound site, facilitating nutrient supply and waste removal, is crucial for collagen deposition, epithelial regeneration, and wound closure. Consequently, enhancing early vascularization and creating a favorable microenvironment for tissue repair have become key areas of research interest. Various hydrogel−based strategies have been evaluated for tissue engineering. While growth factors with pro−angiogenic properties have potential clinical applications, their high cost and instability pose significant limitations. In addition, novel hydrogels modified with nanomaterials, especially magnetic nanoparticles (MNPs), offer many advantages, such as improved mechanical and biological properties [41,42]. However, future research should focus on developing more efficient preparation methods and addressing the challenges associated with optimizing the magnetic responsiveness and biocompatibility [43]. LC materials possess desirable properties such as accessibility, stability, and low toxicity, making them promising biomaterials. LCs have shown potential in improving angiogenesis by facilitating the formation of endothelial cell rings. However, the direct utilization of liquid crystals poses challenges. To address this, a novel biomimetic material was developed by combining the mechanical properties and mesoporous structure of MSN with LCs.

In this study, the LC−MSN composite system was successfully prepared. It was reported that mesoporous materials have a pore size distribution in the range of 2–50 nm [44]. The N_2_ adsorption/desorption isotherms of the MSN synthesized in this experiment showed a typical type IV isotherm, indicating its mesoporosity. The calculations demonstrated that the MSN possessed a large surface area and porosity, enabling high drug−carrying capacity. Following LC modification, the surface area and porosity slightly decreased, suggesting that some LC molecules occupied interstitial spaces without replacing the Si sites in the loose nanoscale network. The stronger electrostatic effect of the LC resulted in a more compact network, leading to a smaller surface area. Chemical structure analysis, optical structure examination using polarized optical microscopy (POM), and crystalline characterization consistently confirmed the retention of liquid crystal properties in the LC−MSN composite. Moreover, the particle size of the synthesized LC−MSN remained stable in the solution, indicating good dispersion and stability without significant aggregation. The hydrogel formed by LC−MSN exhibited favorable biodegradability, meeting the requirements of biomaterials. Collectively, the characterization results confirmed that the synthesized LC−MSN composite aligned with the intended design.

In our study, HUVECs were utilized as the cell models due to their crucial role in angiogenesis, the process of forming tubular structures resembling blood vessels. HUVECs are widely used for studying their biological behavior and response to various stimuli, and they have been extensively employed to investigate the mechanisms involved in angiogenesis [45]. Since we prepared LC−MSN as a new biomaterial, our first priority was to verify its biocompatibility. Cell proliferation and viability studies were conducted to evaluate the cytocompatibility of LC−MSN at various concentrations for up to 48 h. The results demonstrated favorable cytocompatibility of LC−MSN at low concentrations. To ensure biocompatibility, concentrations over 37.5 μg/mL were excluded from subsequent experiments. Notably, when HUVECs were cultured with LC−MSN at a concentration of 18.8 μg/mL, we observed significant promotion of cell extension and migration. These findings provide further evidence of the bioactive environment created by LC−MSN. The ability of endothelial cells to form tubular structures is crucial for angiogenesis, so we also assessed the potential of LC−MSN to promote vascularization using an in vitro tube formation assay. The results revealed that HUVECs cultured with LC−MSN formed more branches, nodes, and complete loops compared to the control group. The enhanced migration ability and actin microfilament performance of HUVECs explain why LC−MSN facilitated the formation of interconnected tubular networks in a shorter time. To further investigate the role of LC−MSN in promoting angiogenesis, qRT−PCR and Western blot analysis were performed. Encouragingly, the expression of several angiogenesis−related genes and proteins, including CD31, VEGF, and ANG, was upregulated in HUVECs cultured with LC−MSN. CD31, a key angiogenic factor, is primarily located at the junctions between endothelial cells and is likely involved in integrin activation for angiogenesis during wound healing. Moreover, angiogenic activity is regulated through complex signaling pathways, with the VEGF−VEGFR and Ang−Tie 2 axes playing significant roles [46]. ANG, acting in conjunction with VEGF, may have a bi−directional impact on angiogenesis. In our study, the increased expression of VEGF suggests that ANG and VEGF may synergistically promote angiogenesis. However, further research is needed to explore the underlying mechanisms.

To take full advantage of its bioactivity, LC−MSN was cross−linked with CS to develop a multifunctional composite hydrogel to promote tissue repair. In vitro experiments demonstrated that the application of the CS/LC−MSN hydrogel significantly reduced the size of wounds compared to the control group. Wound healing is a dynamic process involving numerous cells and bioactive signaling molecules [47]. It comprises sequential and overlapping stages, including hemostasis, inflammation, proliferation, and remodeling. During the proliferation stage, resident cells such as fibroblasts and endothelial cells migrate to the wound site under the influence of host−mediated inflammatory responses [48]. However, the proliferation process is susceptible to interference from excessive inflammation [49]. The in vivo study revealed that CS/LC−MSN modulated the inflammatory response during the inflammation stage by downregulating the expression of IL−1β and IL−6, thereby promoting the transition of wound repair towards the proliferation stage. Key indicators of wound healing during this stage involve epithelial cell regeneration and collagen deposition. The evaluation of healed tissues through H&E−staining and Masson’s staining demonstrated that the CS/LC−MSN hydrogel not only facilitated epithelial regeneration and collagen synthesis but also promoted the formation of dermal appendages, thus approximating the morphological characteristics of normal skin. In line with the results of qPCR and Western blot analysis in vitro, areas with more new vessels were identified in the CS/LC−MSN groups in vivo via a CD31 staining assessment. In summary, these promising findings can be attributed to the favorable moist environment provided by the hydrogel for wound healing, along with the active release of LC−MSN that promotes angiogenesis, collectively creating a conducive microenvironment.

Further studies are needed to investigate the potential mechanisms underlying the observations made in this study. It is also essential to optimize the hydrogel formulation to improve its clinical translation, taking into account long−term safety and potential toxicity concerns. Furthermore, exploring other potential applications of the LC−MSN system, such as its bio−mineralization and osteogenic properties, could further expand its scope of use in tissue engineering.

## 5. Conclusions

In conclusion, we successfully synthesized LC−MSN, which exhibited excellent cytocompatibility and angiogenesis potential. The LC−MSN was further incorporated into a hydrogel to create a multifunctional biomaterial. Through FTIR, 1H NMR, DLS, BET, and POM analyses, we confirmed the successful modification of LC onto MSN. In vitro cell behavior experiments demonstrated the biocompatibility of LC−MSN. Moreover, the biomaterial showed remarkable wound healing potential in terms of in vitro angiogenesis assays. The LC−MSN−incorporated hydrogel not only possessed the inherent advantages of hydrogels but also provided a biomimetic microenvironment due to the presence of LC. In vivo, the biodegradable hybrid hydrogel promoted the formation of granulation tissue, collagen deposition, and neovascularization, thereby enhancing early−stage full−thickness wound healing. Overall, we have great confidence that our study on the LC−MSN−incorporated hydrogel will contribute valuable insights into the development of angiogenic strategies for soft tissue diseases where angiogenesis plays a crucial role.

## Data Availability

Data are available from the corresponding author upon request.

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
