# Peer review of "Biomimetic Liquid Crystal-Modified Mesoporous Silica−Based Composite Hydrogel for Soft Tissue Repair"

_jfb, 2023, doi:10.3390/jfb14060316_

Round 1

Reviewer 1 Report

After reviewing the manuscript entitled: "Biomimetic liquid crystal-modified mesoporous silica-based composite hydrogel for soft tissue repair", from my point of view the context is written well enough to be considered for publication. The authors demonstrate a convenient and effective method to modify the MSN with bioactive LC for improving the angiogenic activity in soft tissue repair. To improve the quality of the manuscript some recommendations are proposed.

1. Addressing the technical/typo suggestions is difficult when the manuscripts don't have line numbers. the authors have used the pronoun "we" several times in the context, which make the context far from the scientific report. Authors should check and re-write the sentences in a passive format. (Specifically in the introduction part, end of page 2!)

2. There are some acronyms in the context without mentioning the complete form like "PBS", "CS", etc. Authors should check all the acronyms and add the complete form the first time it comes in the context.

3. The state of the art is limited. For instance, recent techniques to investigate hydrogel properties and drug delivery such as Molecular Dynamics are missing. The authors should better elaborate their literature survey. Check and refer to these two recent manuscripts.(https://doi.org/10.1016/j.enganabound.2023.02.055) (https://doi.org/10.1016/j.mtcomm.2022.103268)  

4. Some of the results were introduced without in-depth discussion and providing sufficient physics (Fig.4 and Fig.5). Authors should check and elaborate.

Minor problems were detected.

Reviewer 2 Report

In this manuscript, the authors have developed biomimetic liquid crystal-modified mesoporous silica-based composite hydrogel for soft tissue repair. This study is interesting, timely, and can be considered for possible publication after a revision. After a careful review, I have found some points that need to be resolved. 

1. In Section 2.3, full sample preparation details for particular characterization should be provided for better understanding.

2. In Fig. 1, TEM and optical images of pristine MSN should be provided for comparison. Also, what is the scale of the optical image?

3. How was the accumulated degradation of Si ions measured?

4. Digital images of the hydrogels with or without LC-MSN should be provided for a better comparison.

5. Is it possible to evaluate the biomineralization of the composite hydrogels?

6. Moroever, composite hydrogel system should be characterized by FTIR and XRD/XPS for their chemical interactions.

Reviewer 3 Report

In the paper entitled “Biomimetic liquid crystal-modified mesoporous silica-based composite hydrogel for soft tissue repair”, the authors reported the design of mesoporous silica nanoparticles modified with liquid crystal for their subsequent incorporation in chitosan hydrogel for tissue repair applications. The in vitro cytocompatibility and wound healing capability of the designed biomaterials was evaluated with promising results. Moreover, in vivo experiments were also performed, showing the potential of the designed scaffolds for soft tissue repair. The proposed concept is innovative and the work is well-conducted. My main concerns are related to the insufficient review of the current state of the art in this topic, the improvable justification of the proposed approach during the introduction and the lack of some references to support the results/reached conclusions. Moreover, a carefull professional review of the used English style (especially in the introduction section) is essential prior to accept the publication of the work in Journal of Functional Biomaterials. Following, I expose some comments and suggestions that could improve the paper and which I would like the authors to address before consider resubmission:

Avoid the definition of abbreviations that are only used once during the abstract (HUVECs).

Introduction: rephrase the first sentence; please (“The oral and maxillofacial region is full of organs and structures of human body”). “Of human body” makes no sense here.

Introduction: “in our latest in vivo study on animals” is redundant.

Introduction: besides describe the advantages of LC and MSN, in the final part of the introduction you should ellaborate a bit more (short paragraph) why you evaluated the incorporation of these elements in a hydrogel. Which are the advantages of hydrogels? Ideal hydrogel formulation for wound dressing applications? (you can see and refer articles where complex binary and ternary hydrogel formulations were proposed for this application).

Synthesis of LC: although you referred a previous work, I suggest highlighting here the main steps of this synthetic procedure.

Section 3.1:

Provide references that justify your associations between FTIR and NMR peaks and the successful synthesis of LC-MSN nanostructures.

You should provide microscopy images of MSN before their modification with LC to show that their size and morphology do not change in a significant way.

You should display your average values with the associated standard deviation (BET surface area, pore diameter, hydrodynamic size, zeta potential, etc.)

The quality of Figure 1 must be improved: resolution, font size very small in same cases, close square of graph in Figure 1B to be consistent, edge legends very close to numbers, increase size scale bar in C, etc.

Section 3.2:

Justify the use of chitosan for your hydrogels. Why this and not other polymer/s?

Figure 2: increase size scale bars and better explain in the legend of the figure the performed experiments. Increase font size in the final graph.

You should briefly describe the formulation of your hydrogels. Chitosan concentration? Concentration of LC-MSN in the hydrogels? Did you evaluate different concentrations of nanostructures? If not, why did you choose the used one?

The degradation profile of LC-MSN hydrogel was in an acceptable range of 45 to 89% during 14 days, which is slightly less than the degradation rate of blank gels (Figure 2)”. But where can we find the degradation rate of blank gels? You should also represent this curve on the graph to check the difference

Section 3.3:

Again, increase font size and resolution in figure 3.

Why HUVECs was the selected cell line?

Discussion:

You mentioned the potential use of growth factors and the key disadvantages of this strategy. In this way, here or in the introduction section, you should also mention other hydrogel-based strategies that have been evaluated for tissue engineering/wound dressing purposes, such as the design of magnetically-responsive biomaterials (see doi: 10.1021/acsnano.0c08253) or the incorporation of key biofunctionalities within hydrogels through the addition of nanoparticles (electrical conductivity, enhanced biomineralization, etc.)

A professional revision of the english style should be performed before resubmission (especially in the introduction section)

Round 2

Reviewer 1 Report

Thank you for the revisions.
Best wishes.

Minor editing of English language required

Reviewer 2 Report

The authors have improved the manuscript as per the comments given by the Reviewer. In my opinion, this manuscript can now be accepted for publication. 

Reviewer 3 Report

Ok